# ALGORITHMIC STABILITY UNLEASHED: GENERALIZATION BOUNDS WITH UNBOUNDED LOSSES

## ABSTRACT

One of the central problems of statistical learning theory is quantifying the generalization ability of learning algorithms within a probabilistic framework. Algorithmic stability is a powerful tool for deriving generalization bounds, however, it typically builds on a critical assumption that losses are bounded. In this paper, we relax this condition to unbounded loss functions with subweibull diameter. This gives new generalization bound for algorithmic stability and also includes existing results of subgaussian and subexponential diameters as specific cases. Our main probabilistic result is a general concentration inequality for subweibull random variables, which may be of independent interest.

## 1 INTRODUCTION

Algorithmic stability has been a topic of growing interest in learning theory. It is a standard theoretic tool to prove the generalization bounds based on the sensitivity of the algorithm to changes in the learning sample, such as leaving one of the data points out or replacing it with a different one. This approach can be traced back to the foundational works of Vapnik & Chervonenkis (1974), which analyzed the generalization bound for the algorithm of hard-margin Support Vector Machine. The ideas of stability were further developed by Rogers & Wagner (1978), Devroye & Wagner (1979a;b), Lugosi & Pawlak (1994) for the $k$-Nearest-Neighbor algorithm, $k$-local algorithms and potential learning rules, respectively. Interesting insights into stability have also been presented by many authors, such as (Kearns & Ron, 1997; Hardt et al., 2016; Gonen & Shalev-Shwartz, 2017; Kuzborskij & Lampert, 2018; Bassily et al., 2020; Liu et al., 2017; Maurer, 2017; Foster et al., 2019; Yuan & Li, 2023), to mention but a few.

Stability arguments are known for only providing in-expectation error bounds. An extensive analysis of various notions of stability and the corresponding (sometimes) high probability generalization bounds are provided in the seminal work (Bousquet & Elisseeff, 2002). To give high probability bounds, McDiarmid's exponential inequality (McDiarmid, 1998) plays an essential role in the analysis. To satisfy the bounded difference condition in McDiarmid's inequality, a popularly used notion of stability allowing high probability upper bounds called uniform stability is introduced in (Bousquet & Elisseeff, 2002). In the context of uniform stability, a series of papers (Feldman & Vondrak, 2018; 2019; Bousquet et al., 2020; Klochkov & Zhivotovskiy, 2021) provide sharper generalization bounds with probabilities. High probability bounds are necessary for inferring generalization when the algorithm is used many times, which is common in practice. Therefore, as compared to the in-expectation ones, high probability bounds are preferred in the study of the generalization performance.

However, the uniform stability implies the boundedness of the loss function, which might narrow the range of application of these results as the generalization analysis of unbounded losses is becoming increasingly important in many situations (Haddouche et al., 2021), such as regularized regression (Kontorovich, 2014), signal processing (Bakhshizadeh et al., 2020), neural networks (Vladimirova et al., 2019), sample bias correction (Dudík et al., 2005), domain adaptation (Cortes & Mohri, 2014; Ben-David et al., 2006; Mansour et al., 2009), boosting (Dasgupta & Long, 2003), and importance-weighting (Cortes et al., 2019; 2021), etc. For a relaxation, Kutin & Niyogi (2012) introduce a notion of "almost-everywhere" stability and proved valuable extensions of McDiarmid's exponential inequality. It is shown in (Kutin & Niyogi, 2012) that the generalization error can still be bounded when the stability of the algorithm happens only on a subset of large measure. This influential result

has been invoked in a number of interesting papers (El-Yaniv & Pechyony, 2006; Shalev-Shwartz et al., 2010; Hush et al., 2007; Mukherjee et al., 2002; Agarwal & Niyogi, 2009; Rubinstein & Simma, 2012; Rakhlin et al., 2005). However, as noted by Kontorovich (2014), the approach of Kutin & Niyogi (2012) entails too restrictive conditions. It is demonstrated in Kontorovich (2014) that the boundedness of loss functions can be dropped at the expense of a stronger notion of stability and a bounded subgaussian diameter of the underlying metric probability space. This fantastic argument is further, recently, improved to subexponential diameter by Maurer & Pontil (2021).

In this work, we move beyond the subgaussian and subexponential diameters and consider the generalization under a much weaker tail assumption, so-called subweibull distribution (Kuchibhotla & Chakrabortty, 2022; Vladimirova et al., 2020). The subweibull distribution includes the subgaussian and subexponential distributions as specific cases and is inducing more and more attention in learning theory due to that it falls under a broad class of heavy-tailed distributions (Zhang & Wei, 2022; Bong & Kuchibhotla, 2023; Li & Liu, 2022; Madden et al., 2020). In this paper, our contributions are two-fold. Firstly, we provide novel concentration inequalities for general functions of independent subweibull random variables. The technical challenge here is that the subweibull distribution is heavy-tailed so the proof method in the related work (Kontorovich, 2014; Maurer & Pontil, 2021) does not hold in this paper. To counter this difficulty, we address it from the perspective of moment inequality. It should be noted that our probabilistic inequalities may be of independent interest. Secondly, we prove high probability generalization bounds for algorithmic stability with unbounded losses. To this end, we define the subweibull diameter of a metric probability space and prove that it can be used to relax the boundedness condition. The heavy tailedness of subweibull distributions also hinders standard proof techniques. With an application to regularized metric regression, our generalization bound extends results in (Kontorovich, 2014; Maurer & Pontil, 2021) to more scenarios.

The paper is organized as follows. We present our main results in Section 2. The preliminaries relevant to our discussion are stated in Section 2.1. The probabilistic inequalities are provided in Section 2.2. Section 2.3 is devoted to provide generalization bounds for algorithmic stability with unbounded losses. We give proofs in Section 3. In the last, Section 4 concludes this paper. The omitted proof and auxiliary lemmas are deferred to the Appendix.

## 2 MAIN RESULTS

In this section, we present the main results.

### 2.1 PRELIMINARIES

This paper considers metric space. A metric probability space $(\mathcal{X}, d, \mu)$ is a measurable space $\mathcal{X}$ whose Borel $\sigma$-algebra is induced by the metric $d$, endowed with the probability measure $\mu$. We use upper-case letters for random variables and lower-case letters for scalars. Let $(\mathcal{X}_i, d_i, \mu_i)$, $i = 1, ..., n$, be a sequence of metric probability spaces. We define the product probability space

$$\mathcal{X}^n = \mathcal{X}_1 \times \mathcal{X}_2 \times ... \times \mathcal{X}_n$$

with the product measure

$$\mu^n = \mu_1 \times \mu_2 \times ... \times \mu_n$$

and $\ell_1$ product metric

$$d^n(x, x') = \sum_{i=1}^{n} d_i(x_i, x'_i), \quad x, x' \in \mathcal{X}^n.$$

We write $X_i \sim \mu_i$ to mean that $X_i$ is an $\mathcal{X}_i$-valued random variable with law $\mu_i$, i.e., $\mathbb{P}(X_i \in A) = \mu_i(A)$ for all Borel $A \subset \mathcal{X}_i$. We use the notation $X_i^j = (X_i, ..., X_j)$ for all sequences. This notation extends naturally to sequences: $X_1^n \sim \mu^n$. A function $\phi : \mathcal{X} \to \mathbb{R}$ is $L$-Lipschitz if

$$|\phi(x) - \phi(x')| \leq Ld(x, x'), \quad x, x' \in \mathcal{X}.$$

We now define subweibull random variables. A real-valued random variable $X$ is said to be subweibull if it has a bounded $\psi_\alpha$-norm. The $\psi_\alpha$-norm of $X$ for any $\alpha > 0$ is defined as

$$\|X\|_{\psi_\alpha} = \inf \left\{ \sigma \in (0, \infty) : \mathbb{E} \exp \left( \left( \frac{|X|}{\sigma} \right)^\alpha \right) \leq 2 \right\}.$$

As shown in (Kuchibhotla & Chakrabortty, 2022; Vladimirova et al., 2020), the subweibull random variable is characterized by the right tail of the Weibull distribution and generalizes sub-Gaussian and sub-exponential distributions. Particularly, when $\alpha = 1$ or $2$, subweibull random variables reduce to subexponential or subgaussian random variables, respectively. It is obvious that the smaller $\alpha$ is, the heavier tail the random variable has. Further, we define the subweibull diameter $\Delta_\alpha(\mathcal{X}_i)$ of the metric probability space $(\mathcal{X}_i, d_i, \mu_i)$ as

$$\Delta_\alpha(\mathcal{X}_i) = \|d_i(X_i, X_i')\|_{\psi_\alpha},$$

where $X_i, X_i' \sim \mu_i$ are independent.

## 2.2 Concentration Inequalities

This section presents our main probabilistic inequalities, which will be used in the next section, generalization bound with unbounded losses for algorithmic stability.

**Theorem 1.** *Let $X_1, ...., X_n$ are independent random variables with values in a measurable space $\mathcal{X}$ and $f : \mathcal{X}^n \to \mathbb{R}$ is a measurable function. Denote $S = f(X_1, ..., X_{i-1}, X_i, X_{i+1}, ..., X_n)$ and $S_i = f(X_1, ..., X_{i-1}, X_i', X_{i+1}, ..., X_n)$, where $(X_1', ..., X_n')$ is an independent copy of $(X_1, ...., X_n)$. Assume moreover that*

$$|S - S_i| \leq F_i(X_i, X_i')$$

*for some functions $F_i : \mathcal{X}^2 \to \mathbb{R}$, $i = 1, ..., n$. Suppose that $\|F_i(X_i, X_i')\|_{\psi_\alpha} < \infty$ for all $i$. Then for any $0 < \delta < 1/e^2$, with probability at least $1 - \delta$*

*(1) if $0 < \alpha \leq 1$, let $c_\alpha = 2((\log 2)^{1/\alpha} + e^3 \Gamma^{1/2}(\frac{2}{\alpha} + 1) + e^3 3^{\frac{2-\alpha}{3\alpha}} \sup_{p \geq 2} p^{\frac{-1}{\alpha}} \Gamma^{1/p}(\frac{p}{\alpha} + 1))$, we have*

$$|f(X_1, ..., X_n) - \mathbb{E}f(X_1, ..., X_n)|$$
$$\leq c_\alpha \left( \sqrt{\log(\frac{1}{\delta})} \left( \sum_{i=1}^n \|F_i(X_i, X_i')\|_{\psi_\alpha}^2 \right)^{\frac{1}{2}} + \log^{1/\alpha}(\frac{1}{\delta}) \max_{1 \leq i \leq n} \|F_i(X_i, X_i')\|_{\psi_\alpha} \right);$$

*(2) if $\alpha > 1$, let $1/\alpha^* + 1/\alpha = 1$ and $c_\alpha' = \max\{8e + 2(\log 2)^{1/\alpha}, 8e(1/\alpha)^{1/\alpha}(1 - \alpha^{-1})^{1/\alpha^*}\}$, and let $(\|F(X, X')\|_{\psi_\alpha}) = (\|F_1(X_1, X_1')\|_{\psi_\alpha}, ..., \|F_n(X_n, X_n')\|_{\psi_\alpha}) \in \mathbb{R}^n$, we have*

$$|f(X_1, ..., X_n) - \mathbb{E}f(X_1, ..., X_n)|$$
$$\leq c_\alpha' \left( \sqrt{\log(\frac{1}{\delta})} \left( \sum_{i=1}^n \|F_i(X_i, X_i')\|_{\psi_\alpha}^2 \right)^{\frac{1}{2}} + \log^{1/\alpha}(\frac{1}{\delta}) \|(\|F(X, X')\|_{\psi_\alpha})\|_{\alpha^*} \right).$$

**Remark 1.** In the context of the subweibull diameter, Theorem 1 is (1) if $0 < \alpha \leq 1$, we have

$$|f(X_1, ..., X_n) - \mathbb{E}f(X_1, ..., X_n)| \leq c_\alpha \left( \sqrt{\log(\frac{1}{\delta})} \left( \sum_{i=1}^n \Delta_\alpha^2(\mathcal{X}_i) \right)^{\frac{1}{2}} + \log^{\frac{1}{\alpha}}(\frac{1}{\delta}) \max_{1 \leq i \leq n} \Delta_\alpha(\mathcal{X}_i) \right);$$

(2) if $\alpha > 1$, let $(\Delta_\alpha(\mathcal{X})) = (\Delta_\alpha(\mathcal{X}_1), ..., \Delta_\alpha(\mathcal{X}_n))$, we have

$$|f(X_1, ..., X_n) - \mathbb{E}f(X_1, ..., X_n)| \leq c_\alpha' \left( \sqrt{\log(\frac{1}{\delta})} \left( \sum_{i=1}^n \Delta_\alpha^2(\mathcal{X}_i) \right)^{\frac{1}{2}} + \log^{\frac{1}{\alpha}}(\frac{1}{\delta}) \|(\Delta_\alpha(\mathcal{X}))\|_{\alpha^*} \right).$$

We now discuss the impact of the value of $\alpha$ on the inequality. For the constant $c_\alpha$, according to the property of Gamma function, $\Gamma(\frac{2}{\alpha} + 1)$ becomes bigger as $\alpha$ becomes smaller. As for the term

$\sup_{p \geq 2} p^{\frac{-1}{\alpha}} \Gamma^{1/p}(\frac{p}{\alpha} + 1)$, the Stirling formula easily gives a concise form that only depends on $\alpha$ (refer to the appendix for details), and we can find that the smaller $\alpha$ is, the bigger this term is. For the constant $c'_\alpha$, we have the inequality $8e + 2(\log 2)^{1/\alpha} \geq 8e(1/\alpha)^{1/\alpha}(1 - \alpha^{-1})^{1/\alpha^*}$, which suggests that the smaller $\alpha$ is, the bigger $c'_\alpha$ is. For the subweibull diameter $\Delta_\alpha(\mathcal{X}_i)$, according to the definition, we know that the smaller $\alpha$ is, the bigger this subweibull diameter is and the heavier tail the random variable has. By the above analysis, we conclude that the smaller the $\alpha$, the bigger the value of the inequality. This result is consistent with the intuition: heavier-tailed distribution, i.e., smaller $\alpha$, will lead to a bigger upper bound.

Let us see how Theorem 1 compares to previous results on some examples. Theorem 1 in (Kontorovich, 2014) states that if $f$ is 1-Lipschitz function, then

$$\mathbb{P}(|f(X_1, ..., X_n) - \mathbb{E}f(X_1, ..., X_n)| > t) \leq 2\exp\left(-\frac{t^2}{2\sum_{i=1}^n \Delta_2^2(\mathcal{X}_i)}\right).$$

Theorem 11 in (Maurer & Pontil, 2021) shows that if $f$ is 1-Lipschitz function, a one-sided inequality holds

$$\mathbb{P}(f(X_1, ..., X_n) - \mathbb{E}f(X_1, ..., X_n) > t) \leq \exp\left(\frac{-t^2}{4e\sum_{i=1}^n \Delta_1^2(\mathcal{X}_i) + 2e\max_i \Delta_1(\mathcal{X}_i)t}\right).$$

By comparison, it is clear that when $\alpha = 2$ or $\alpha = 1$, our inequalities, respectively, reduce to the ones in Kontorovich (2014); Maurer & Pontil (2021), respectively, up to constants. Our inequalities are two-sided compared to the one in (Maurer & Pontil, 2021). As for proof techniques, the goal of McDiarmid's inequality is to deal with the concentration of the general function $f$. Related work Kontorovich (2014) used the martingale method to decompose $f - \mathbb{E}f$, while Maurer & Pontil (2021) use the sub-additivity of entropy to decompose the general function $f - \mathbb{E}f$. After the decomposition, the next step of Kontorovich (2014) and Maurer & Pontil (2021) is to bound the moment generating function (MGF) ($\mathbb{E}e^{\lambda X}$) or a variant MGF ($\mathbb{E}Z^2 e^{\lambda X}$), respectively. The MGF is bounded for sub-Gaussian and sub-exponential random variables, however it is unbounded for subweibull variables because there is some convexity lost. The standard technique to prove the MGF failed for the heavy-tailed subweibull random variables. This implies that if we do not study the MGF, we need to consider different decomposition on the general function $f - \mathbb{E}f$. To counter this difficulty, we address it from the perspective of $p$-th moment inequality via an induction approach. We introduce a technical lemma, i.e., Lemma 4. In the proof of Lemma 4, a key step is that we need to construct a function $t \to h(\epsilon_n F_n(X_n, X'_n) + t)$ to apply our induction assumption. This Lemma is very useful. For example, one can use Lemma 4 to prove more McDiarmid-type inequalities, e.g., the polynomially decaying random variables, which will enrich the family of McDiarmid-type inequalities. With Lemma 4, we firstly decompose the concentration of the general function $f$ to the sum of independent subweibull random variables. Rather than bounding the MGF, we then bound the $p$-th moment of the sum of subweibull random variables, refer to Lemma 5. Thanks to the fact that subweibull random variables are log-convex for $\alpha \leq 1$ and log-concave for $\alpha \geq 1$, we can apply Latała's inequality (Lemma 7 and Lemma 8) to bound this $p$-th moment. However, it is not a direct application of the Latała's inequality. Latała's inequality holds for the $p$-th moment of the *symmetric* random variables. On one hand, we need to carefully construct new random variables to satisfy the symmetry condition, for example, we introduce random variables $Y_i$, $Z_i$ in the proof of Lemma 5. On the other hand, since we study the weighted summation, Khinchin-Kahane inequality is also required to use. Please refer to the proof in Section 3 for more details.

## 2.3 ALGORITHMIC STABILITY WITH UNBOUNDED LOSSES

In this section, the metric probability space $(\mathcal{Z}_i, d_i, \mu_i)$ will have the structure $\mathcal{Z}_i = \mathcal{X}_i \times \mathcal{Y}_i$ where $\mathcal{X}_i$ and $\mathcal{Y}_i$ are the instance and label space of the $i$-th example, respectively. Under the i.i.d assumption, the $(\mathcal{Z}_i, d_i, \mu_i)$ are identical for all $i \in \mathbb{N}$, and so we will henceforth drop the subscript $i$ from these. We are given an i.i.d sample of points $S = Z_1^n \sim \mu^n$, and a learning algorithm $\mathcal{A} : (\mathcal{X} \times \mathcal{Y})^n \to \mathcal{Y}^{\mathcal{X}}$ maps a training sample to a function mapping the instance space $\mathcal{X}$ into the space of labels $\mathcal{Y}$. The output of the learning algorithm based on the sample $S$ will be denoted by $\mathcal{A}_S$. The quality of the function returned by the algorithm is measured using a loss function $\ell : \mathcal{Y} \times \mathcal{Y} \to \mathbb{R}_+$. The empirical risk $R_n(\mathcal{A}, S)$ is typically defined as

$$R_n(\mathcal{A}, S) = \frac{1}{n}\sum_{i=1}^n \ell(\mathcal{A}_S, z_i)$$

and the population risk $R(\mathcal{A}, S)$ as

$$R(\mathcal{A}, S) = \mathbb{E}_{z \sim \mu}[\ell(\mathcal{A}_S, z)].$$

One of the fundamental questions in statistical learning is to estimate the risk $R(\mathcal{A}, S)$ of an algorithm from the sample $S$ in terms of the empirical one. A large body of work has been dedicated to obtaining generalization bounds, i.e., high probability bounds on the error of the empirical risk estimator: $R(\mathcal{A}, S) - R_n(\mathcal{A}, S)$. The widely used notion of stability allowing high probability upper bounds is called uniform stability. We mention a variant of uniform stability provided in (Rakhlin et al., 2005), which is slightly more general than the original notion in (Bousquet & Elisseeff, 2002). The algorithm $A$ is said to be $\gamma$-uniform stable if for any $\bar{z} \in \mathcal{Z}$, the function $\psi : \mathcal{Z}^n \to \mathbb{R}$ given by $\psi(z_1^n) = \ell(\mathcal{A}_{z_1^n}, \bar{z})$ is $\gamma$-Lipschitz with respect to the Hamming metric on $\mathcal{Z}^n$:

$$\forall z, z' \in \mathcal{Z}^n, \forall \bar{z} \in \mathcal{Z} : |\psi(z) - \psi(z')| \leq \gamma \sum_{i=1}^{n} \mathbb{I}_{\{z_i \neq z_i'\}}.$$

Most previous work of stability required the loss to be bounded by some constant $M < \infty$. We make no such restriction in this paper. To relax the boundedness condition, we use a different notion of stability proposed in (Kontorovich, 2014). Specifically, the algorithm $\mathcal{A}$ is said to be $\gamma$-totally Lipschitz stable if the function $\psi : \mathcal{Z}^{n+1} \to \mathbb{R}$ given by $\psi(z_1^{n+1}) = \ell(\mathcal{A}_{z_1^n}, z_{n+1})$ is $\gamma$-Lipschitz with respect to the $\ell_1$ product metric on $\mathcal{Z}^{n+1}$:

$$\forall z, z' \in \mathcal{Z}^{n+1} : |\psi(z) - \psi(z')| \leq \gamma \sum_{i=1}^{n+1} d(z_i, z_i').$$

We now give a generalization bound of stable algorithms.

**Lemma 1.** *Suppose $\mathcal{A}$ is a symmetric, $\gamma$-totally Lipschitz stable learning algorithm over the metric probability space $(\mathcal{Z}, d, \mu)$ with $\Delta_\alpha(\mathcal{Z}) < \infty$. Then*

$$\mathbb{E}[R(\mathcal{A}, S) - R_n(\mathcal{A}, S)] \leq c(\alpha)\gamma \Delta_\alpha(\mathcal{Z}),$$

*where $c(\alpha) = (\log 2)^{1/\alpha}$ if $\alpha > 1$ and $c(\alpha) = 2\Gamma(\frac{1}{\alpha} + 1)$ if $0 < \alpha \leq 1$.*

**Remark 2.** This proof is delayed to the Appendix. The heavy tailedness of subweibull distributions hinders standard proof techniques, such as the Jensen's inequality.

The next lemma discusses Lipschitz continuity.

**Lemma 2** (Lemma 2 in (Kontorovich, 2014)). *Suppose $\mathcal{A}$ is a symmetric, $\gamma$-totally Lipschitz stable learning algorithm and define the function $f : \mathcal{Z}^n \to \mathbb{R}$ by $f(z) = R(\mathcal{A}, z) - R_n(\mathcal{A}, z)$. Then $f$ is $3\gamma$-Lipschitz.*

Combining Lemma 2 with Theorem 1 and together with Lemma 1 yields the following high probability generalization bound.

**Theorem 2.** *Suppose $\mathcal{A}$ is a symmetric, $\gamma$-totally Lipschitz stable learning algorithm over the metric probability space $(\mathcal{Z}, d, \mu)$ with $\Delta_\alpha(\mathcal{Z}) < \infty$. Then, for training samples $S \sim \mu^n$ and any $0 < \delta < 1/e^2$, with probability at least $1 - \delta$*

*(1) if $0 < \alpha \leq 1$, let $c_\alpha = 2\sqrt{2}((\log 2)^{1/\alpha} + e^3 \Gamma^{1/2}(\frac{2}{\alpha} + 1) + e^3 3^{\frac{2-\alpha}{3\alpha}} \sup_{p \geq 2} p^{\frac{-1}{\alpha}} \Gamma^{1/p}(\frac{p}{\alpha} + 1))$, we have*

$$R(\mathcal{A}, S) - R_n(\mathcal{A}, S) \leq c(\alpha)\gamma \Delta_\alpha(\mathcal{Z}) + 3\gamma c_\alpha \left( \sqrt{n \log(\frac{1}{\delta})} \Delta_\alpha(\mathcal{Z}) + \log^{\frac{1}{\alpha}}(\frac{1}{\delta}) \Delta_\alpha(\mathcal{Z}) \right),$$

*(2) if $\alpha > 1$, let $1/\alpha^* + 1/\alpha = 1$ and $c_\alpha' = \max\{8e + 2(\log 2)^{1/\alpha}, 8e(1/\alpha)^{1/\alpha}(1 - \alpha^{-1})^{1/\alpha^*}\}$, we have*

$$R(\mathcal{A}, S) - R_n(\mathcal{A}, S) \leq c(\alpha)\gamma \Delta_\alpha(\mathcal{Z}) + 3\gamma c_\alpha' \left( \sqrt{n \log(\frac{1}{\delta})} \Delta_\alpha(\mathcal{Z}) + \log^{\frac{1}{\alpha}}(\frac{1}{\delta}) n^{\frac{1}{\alpha^*}} \Delta_\alpha(\mathcal{Z}) \right),$$

*where $c(\alpha) = (\log 2)^{1/\alpha}$ if $\alpha > 1$ and $c(\alpha) = 2\Gamma(\frac{1}{\alpha} + 1)$ if $0 < \alpha \leq 1$.*

**Remark 3.** Similar to the discussion in Remark 1, the relationship between $\alpha$ and the generalization in Theorem 2 is that a heavier-tailed random variable, i.e., a smaller $\alpha$, results in a bigger bound, i.e., a looser generalization bound. Next, we compare Theorem 2 with previous results. For a pair of non-negative functions $f, g$ the notation $f \lesssim g$ will mean that for some universal constant $c > 0$ it holds that $f \leq cg$. In the related work, the basic and the best known result is the high probability upper bound in (Bousquet & Elisseeff, 2002) which states that, with probability at least $1 - \delta$,

$$R(\mathcal{A}, S) - R_n(\mathcal{A}, S) \lesssim \left(\gamma^2 + \frac{M}{\sqrt{n}}\right)\sqrt{\log(\frac{1}{\delta})},$$

where $\gamma$ denotes the uniform stability and $M$ is the upper bound of the loss $\ell$. Kontorovich (2014) extends this bound to the unbounded loss with subgaussian diameter, and their Theorem 2 states that, with probability at least $1 - \delta$,

$$R(\mathcal{A}, S) - R_n(\mathcal{A}, S) \lesssim \gamma^2 \Delta_2^2(\mathcal{Z}) + \gamma\Delta_2(\mathcal{Z})\sqrt{n\log(\frac{1}{\delta})},$$

where, in this case, $\gamma$ denotes the totally Lipschitz stability and $\Delta_2(\mathcal{Z})$ is the subgaussian diameter. If we instead consider the subexponential distributions, the generalization bound in (Maurer & Pontil, 2021) is

$$R(\mathcal{A}, S) - R_n(\mathcal{A}, S) \lesssim \gamma\Delta_1(\mathcal{Z}) + \gamma\Delta_1(\mathcal{Z})\sqrt{n\log(\frac{1}{\delta})} + \gamma\Delta_1(\mathcal{Z})\log(\frac{1}{\delta}).$$

As shown in the above three bounds and related results on algorithmic stability, the stability $\gamma$ is required at the least of the order $1/\sqrt{n}$ for nontrivial convergence decay. By comparison to the relevant bounds in (Bousquet & Elisseeff, 2002; Kontorovich, 2014; Maurer & Pontil, 2021), our generalization bounds in Theorem 2 give results for unbounded loss functions with subweibull diameter, which includes the results of (Kontorovich, 2014; Maurer & Pontil, 2021) as specific cases and substantially extends the existing results to a large broad class of unbounded losses. Recently, Yuan & Li (2023) provide sharper high probability generalization bounds for unbounded losses up to subexponential distributions (i.e., $\alpha = 1$) in the sense of (Bousquet et al., 2020; Klochkov & Zhivotovskiy, 2021). As a comparison, our generalization bounds allow heavy-tailed distributions (i.e., $0 < \alpha < 1$). It would be interesting to study whether our techniques can be used to give sharper bounds for heavy-tailed subweibull distributions in the spirit of (Bousquet et al., 2020; Klochkov & Zhivotovskiy, 2021; Yuan & Li, 2023).

**Remark 4** (Application to Regularized Metric Regression). We first give some necessary notations of the regularized regression. We assume the label space $\mathcal{Y}$ to be all of $\mathbb{R}$. A simple no-free-lunch argument shows that it is impossible to learn functions with arbitrary oscillation, and hence Lipschitzness is a natural and commonly used regularization constraint (Shalev-Shwartz & Ben-David, 2014; Tsybakov, 2003; Wasserman, 2006). We will denote by $\mathcal{F}_\lambda$ the collection of all $\lambda$-Lipschitz functions $f : \mathcal{X} \to \mathbb{R}$. The learning algorithm $\mathcal{A}$ maps the sample $S = Z_{i=1}^n$, with $Z_i = (X_i, Y_i) \in \mathcal{X} \times \mathbb{R}$, to the function $\hat{f} \in \mathcal{F}_\lambda$ by minimizing the empirical risk

$$\hat{f} = \arg\min_{f \in \mathcal{F}_\lambda} \frac{1}{n}\sum_{i=1}^n |f(X_i) - Y_i|$$

over all $f \in \mathcal{F}_\lambda$, where we have chosen the absolute loss $\ell(y, y') = |y - y'|$. In the general metric space, Gottlieb et al. (2017) proposed an efficient algorithm for regression via Lipschitz extension, a method that can be traced back to the seminal work (von Luxburg & Bousquet, 2004), which is algorithmically realized by 1-nearest neighbors. This approach very facilitates generalization analysis. For any metric space $(\mathcal{X}, d)$, we associate it to a metric space $(\mathcal{Z}, \bar{d})$, where $\mathcal{Z} = \mathcal{X} \times \mathbb{R}$ and $\bar{d}((x, y), (x', y')) = d(x, x') + |y - y'|$, and we suppose that $(\mathcal{Z}, \bar{d})$ is endowed with a measure $\mu$ such that $\Delta_\alpha(\mathcal{Z}) = \Delta_\alpha(\mathcal{Z}, \bar{d}, \mu) < \infty$.

Here, the standard order of magnitude notation such as $O(\cdot)$ and $\Omega(\cdot)$ will be used. If none of the $n + 1$ points ($n$ sample and 1 test) is too isolated from the rest, Kontorovich (2014) shows that the regression algorithm is $\gamma = O(\lambda/n)$-totally Lipschitz stable. In the case of subgaussian distribution, with probability $1 - n\exp(-\Omega(n))$, each of the $n+1$ points is within distance $O(\Delta_2(\mathcal{Z}))$ of another point. Hence, Kontorovich (2014) states that, with probability at least $1 - n\exp(-\Omega(n)) - \delta$

$$R(\mathcal{A}, S) - R_n(\mathcal{A}, S) \lesssim \left(\frac{\lambda}{n}\Delta_2(\mathcal{Z})\right)^2 + \frac{\lambda}{\sqrt{n}}\Delta_2(\mathcal{Z})\sqrt{\log(\frac{1}{\delta})}.$$

While in the case of subweibull distribution, according to Theorem 2.1 in (Vladimirova et al., 2020), with probability $1 - n\exp(-\Omega(n^\alpha))$, each of the $n + 1$ points is within distance $O(\Delta_\alpha(\mathcal{Z}))$ of another point. Thus, by Theorem 2, our bound is, with probability at least $1 - n\exp(-\Omega(n^\alpha)) - \delta$, (1) if $0 < \alpha \leq 1$,

$$R(\mathcal{A}, S) - R_n(\mathcal{A}, S) \lesssim \frac{\lambda}{n}\Delta_\alpha(\mathcal{Z}) + \frac{\lambda}{n}\left(\sqrt{n\log(\frac{1}{\delta})}\Delta_\alpha(\mathcal{Z}) + \log^{\frac{1}{\alpha}}(\frac{1}{\delta})\Delta_\alpha(\mathcal{Z})\right);$$

(2) if $\alpha > 1$, let $1/\alpha^* + 1/\alpha = 1$,

$$R(\mathcal{A}, S) - R_n(\mathcal{A}, S) \lesssim \frac{\lambda}{n}\Delta_\alpha(\mathcal{Z}) + \frac{\lambda}{n}\left(\sqrt{n\log(\frac{1}{\delta})}\Delta_\alpha(\mathcal{Z}) + \log^{\frac{1}{\alpha}}(\frac{1}{\delta})n^{\frac{1}{\alpha^*}}\Delta_\alpha(\mathcal{Z})\right).$$

As a comparison, our results allow a substantial extension of existing generalization bounds to heavy-tailed distributions.

## 3 PROOFS

This section proves Theorem 1. To proceed, we state three technical lemmas.

**Lemma 3.** *Let $h : \mathbb{R} \to \mathbb{R}$ be a convex functions, $\epsilon_1, ...., \epsilon_n$ a sequence of independent Rademacher variables and $a_1, ..., a_n$, $b_1, ..., b_n$ two sequences of nonnegative real numbers, such that for every $i$ $a_i \leq b_i$. Then*

$$\mathbb{E}h\left(\sum_{i=1}^n a_i\epsilon_i\right) \leq \mathbb{E}h\left(\sum_{i=1}^n b_i\epsilon_i\right).$$

**Lemma 4.** *Let $h : \mathbb{R} \to \mathbb{R}$ be a convex function and $S = f(X_1, ..., X_{i-1}, X_i, X_{i+1}, ..., X_n)$, where $X_1, ...., X_n$ are independent random variables with values in a measurable space $\mathcal{X}$ and $f : \mathcal{X}^n \to \mathbb{R}$ is a measurable function. Denote as usual $S_i = f(X_1, ..., X_{i-1}, X_i', X_{i+1}, ..., X_n)$, where $(X_1', ..., X_n')$ is an independent copy of $(X_1, ...., X_n)$. Assume moreover that $|S - S_i| \leq F_i(X_i, X_i')$ for some functions $F_i : \mathcal{X}^2 \to \mathbb{R}$, $i = 1, ..., n$. Then,*

$$\mathbb{E}h(S - \mathbb{E}S) \leq \mathbb{E}h\left(\sum_{i=1}^n \epsilon_i F_i(X_i, X_i')\right),$$

*where $\epsilon_1, ..., \epsilon_n$ is a sequence of independent Rademacher variables, independent of $(X_i)_{i=1}^n$ and $(X_i')_{i=1}^n$.*

Next lemma provides a moment inequality for the sum of independent subweibull random variables.

**Lemma 5.** *Suppose $X_1, X_2, ..., X_n$ are independent subweibull random variables with mean zero. For any vector $a = (a_1, ..., a_n) \in \mathbb{R}^n$, let $b = (a_1\|X_1\|_{\psi_\alpha}, ..., a_n\|X_n\|_{\psi_\alpha}) \in \mathbb{R}^n$. Then for $p \geq 1$,*

- *if $0 < \alpha \leq 1$, let $c_\alpha = 2\sqrt{2}((\log 2)^{1/\alpha} + e^3\Gamma^{1/2}(\frac{2}{\alpha}+1) + e^3 3^{\frac{2-\alpha}{3\alpha}}\sup_{p\geq 2} p^{\frac{-1}{\alpha}}\Gamma^{1/p}(\frac{p}{\alpha}+1))$,*

$$\left\|\sum_{i=1}^n a_i X_i\right\|_p \leq c_\alpha\left(\sqrt{p}\|b\|_2 + p^{1/\alpha}\|b\|_\infty\right).$$

- *if $\alpha > 1$, let $1/\alpha^* + 1/\alpha = 1$ and $c_\alpha' = \max\{8e + 2(\log 2)^{1/\alpha}, 8e(1/\alpha)^{1/\alpha}(1-\alpha^{-1})^{1/\alpha^*}\}$,*

$$\left\|\sum_{i=1}^n a_i X_i\right\|_p \leq c_\alpha'\left(\sqrt{p}\|b\|_2 + p^{1/\alpha}\|b\|_{\alpha^*}\right).$$

*Proof.* Without loss of generality, we assume $\|X_i\|_{\psi_\alpha} = 1$. Define $Y_i = (|X_i| - (\log 2)^{1/\alpha})_+$, then it is easy to check that $\mathbb{P}(|X_i| \geq t) \leq 2e^{-t^\alpha}$, which also implies that $\mathbb{P}(Y_i \geq t) \leq e^{-t^\alpha}$. According to the symmetrization inequality (e.g., Proposition 6.3 of (Ledoux & Talagrand, 1991)), we have

$$\|\sum_{i=1}^n a_i X_i\|_p \leq 2\|\sum_{i=1}^n \epsilon_i a_i X_i\|_p = 2\|\sum_{i=1}^n \epsilon_i a_i |X_i|\|_p,$$

where $\{\epsilon_i\}_{i=1}^n$ are independent Rademacher random variables and we have used that $\epsilon_i X_i$ and $\epsilon_i |X_i|$ are identically distributed. By triangle inequality,

$$2\|\sum_{i=1}^n \epsilon_i a_i |X_i|\|_p \le 2\|\sum_{i=1}^n \epsilon_i a_i (Y_i + (\log 2)^{1/\alpha})\|_p \le 2\|\sum_{i=1}^n \epsilon_i a_i Y_i\|_p + 2(\log 2)^{1/\alpha}\|\sum_{i=1}^n \epsilon_i a_i\|_p.$$

Next, we will bound the second term of the RHS of the above bound. By Khinchin-Kahane inequality (Lemma 6 in the Appendix), we have

$$\|\sum_{i=1}^n \epsilon_i a_i\|_p \le (\frac{p-1}{2-1})^{1/2}\|\sum_{i=1}^n \epsilon_i a_i\|_2 \le \sqrt{p}\|\sum_{i=1}^n \epsilon_i a_i\|_2 = \sqrt{p}(\mathbb{E}(\sum_{i=1}^n \epsilon_i a_i)^2)^{1/2}$$

$$= \sqrt{p}\big(\mathbb{E}(\sum_{i=1}^n \epsilon_i^2 a_i^2 + 2\sum_{1 \le i < j \le n} \epsilon_i \epsilon_j a_i a_j)\big)^{1/2} = \sqrt{p}(\sum_{i=1}^n a_i^2)^{1/2} = \sqrt{p}\|a\|_2.$$

Let $\{Z_i\}_{i=1}^n$ be independent symmetric random variables satisfying $\mathbb{P}(|Z_i| \ge t) = \exp(-t^\alpha)$ for all $t \ge 0$, we have

$$\|\sum_{i=1}^n \epsilon_i a_i Y_i\|_p \le \|\sum_{i=1}^n \epsilon_i a_i Z_i\|_p = \|\sum_{i=1}^n a_i Z_i\|_p,$$

since $\epsilon_i Z_i$ and $Z_i$ have the same distribution due to symmetry. Combining the above inequalities together, we reach

$$\|\sum_{i=1}^n a_i X_i\|_p \le 2(\log 2)^{1/\alpha}\sqrt{p}\|a\|_2 + 2\|\sum_{i=1}^n a_i Z_i\|_p.$$

In the case of $0 < \alpha \le 1$, $N(t) = t^\alpha$ is concave. Then Lemma 7 and Lemma 8 (a) (in the Appendix) gives for $p \ge 2$

$$\|\sum_{i=1}^n a_i Z_i\|_p \le e \inf\{t > 0 : \sum_{i=1}^n \log \psi_p(e^{-2}(\frac{a_i e^2}{t})Z_i) \le p\} \le e \inf\{t > 0 : \sum_{i=1}^n p M_{p,Z_i}(\frac{a_i e^2}{t}) \le p\}$$

$$= e \inf\{t > 0 : \sum_{i=1}^n (\max\{(\frac{a_i e^2}{t})^p \|Z_i\|_p^p, p(\frac{a_i e^2}{t})^2 \|Z_i\|_2^2\}) \le p\}$$

$$\le e \inf\{t > 0 : \sum_{i=1}^n (\frac{a_i e^2}{t})^p \|Z_i\|_p^p + \sum_{i=1}^n p(\frac{a_i e^2}{t})^2 \|Z_i\|_2^2 \le p\}$$

$$\le e \inf\{t > 0 : 2p\Gamma(\frac{p}{\alpha} + 1)\frac{e^{2p}}{t^p}\|a\|_p^p \le p\} + e \inf\{t > 0 : 2p^2\Gamma(\frac{2}{\alpha} + 1)\frac{e^4}{t^2}\|a\|_2^2] \le p\},$$

where we have used $\|Z_i\|_p^p = p\Gamma(\frac{p}{\alpha} + 1)$. Thus,

$$\|\sum_{i=1}^n a_i Z_i\|_p \le \sqrt{2}e^3(\Gamma^{1/p}(\frac{p}{\alpha} + 1)\|a\|_p + \sqrt{p}\Gamma^{1/2}(\frac{2}{\alpha} + 1)\|a\|_2).$$

By homogeneity, we can assume that $\sqrt{p}\|a\|_2 + p^{1/\alpha}\|a\|_\infty = 1$. Then $\|a\|_2 \le p^{-1/2}$ and $\|a\|_\infty \le p^{-1/\alpha}$. Therefore, for $p \ge 2$,

$$\|a\|_p \le (\sum_{i=1}^n |a_i|^2 \|a\|_\infty^{p-2})^{1/p} \le (p^{-1-(p-2)/\alpha})^{1/p} = (p^{-p/\alpha}p^{(2-\alpha)/\alpha})^{1/p} \le 3^{\frac{2-\alpha}{3\alpha}}p^{-1/\alpha}$$

$$= 3^{\frac{2-\alpha}{3\alpha}}p^{-1/\alpha}(\sqrt{p}\|a\|_2 + p^{1/\alpha}\|a\|_\infty),$$

where we used $p^{1/p} \le 3^{1/3}$ for any $p \ge 2, p \in \mathbb{N}$. Therefore, for $p \ge 2$,

$$\|\sum_{i=1}^n a_i X_i\|_p \le 2(\log 2)^{1/\alpha}\sqrt{p}\|a\|_2 + 2\sqrt{2}e^3(\Gamma^{1/p}(\frac{p}{\alpha} + 1)\|a\|_p + \sqrt{p}\Gamma^{1/2}(\frac{2}{\alpha} + 1)\|a\|_2)$$

$$\le 2\sqrt{2}((\log 2)^{1/\alpha} + e^3\Gamma^{1/2}(\frac{2}{\alpha} + 1) + e^3 3^{\frac{2-\alpha}{3\alpha}}p^{\frac{-1}{\alpha}}\Gamma^{1/p}(\frac{p}{\alpha} + 1))\sqrt{p}\|a\|_2$$

$$+ 2\sqrt{2}e^{3+\frac{2-\alpha}{e\alpha}}\Gamma^{1/p}(\frac{p}{\alpha} + 1)\|a\|_\infty.$$

Let $c_\alpha = 2\sqrt{2}((\log 2)^{1/\alpha} + e^3\Gamma^{1/2}(\frac{2}{\alpha} + 1) + e^3 3^{\frac{2-\alpha}{3\alpha}} \sup_{p \geq 2} p^{\frac{-1}{\alpha}}\Gamma^{1/p}(\frac{p}{\alpha} + 1))$, we have

$$\|\sum_{i=1}^{n} a_i X_i\|_p \leq c_\alpha(\sqrt{p}\|a\|_2 + p^{1/\alpha}\|a\|_\infty).$$

In the case of $\alpha > 1$, $N(t) = t^\alpha$ is convex with $N^*(t) = \alpha^{-\frac{1}{\alpha-1}}(1 - \alpha^{-1})t^{\frac{\alpha}{\alpha-1}}$. Then Lemma 7 and Lemma 7 (b) (in the Appendix) gives for $p \geq 2$

$$\|\sum_{i=1}^{n} a_i Z_i\|_p \leq e\inf\{t > 0 : \sum_{i=1}^{n}\log\psi_p(\frac{4a_i}{t}Z_i/4) \leq p\} + e\inf\{t > 0 : \sum_{i=1}^{n} pM_{p,Z_i}(\frac{4a_i}{t}) \leq p\}$$

$$\leq e\inf\{t > 0 : \sum_{i=1}^{n} p^{-1}N^*(p|\frac{4a_i}{t}|) \leq 1\} + e\inf\{t > 0 : \sum_{i=1}^{n} p(\frac{4a_i}{t})^2 \leq 1\}$$

$$= 4e(\sqrt{p}\|a\|_2 + (p/\alpha)^{1/\alpha}(1 - \alpha^{-1})^{1/\alpha^*}\|a\|_{\alpha^*}),$$

where $\alpha^*$ is mentioned in the statement. Therefore, for $p \geq 2$,

$$\|\sum_{i=1}^{n} a_i X_i\|_p \leq (8e + 2(\log 2)^{1/\alpha})\sqrt{p}\|a\|_2 + 8e(1/\alpha)^{1/\alpha}(1 - \alpha^{-1})^{1/\alpha^*}p^{1/\alpha}\|a\|_{\alpha^*}.$$

Let $c'_\alpha = \max\{8e + 2(\log 2)^{1/\alpha}, 8e(1/\alpha)^{1/\alpha}(1 - \alpha^{-1})^{1/\alpha^*}\}$, we have

$$\|\sum_{i=1}^{n} a_i X_i\|_p \leq c'_\alpha(\sqrt{p}\|a\|_2 + p^{1/\alpha}\|a\|_{\alpha^*}).$$

Replacing $a$ with $b$, the proof is complete. $\qquad\square$

*Proof of Theorem 1.* Using Lemma 4 with $h(t) = |t|^p$, for $p \geq 2$,

$$\|f(X_1, ..., X_n) - \mathbb{E}f(X_1, ..., X_n)\|_p \leq \left\|\sum_{i=1}^{n} \epsilon_i F_i(X_i, X'_i)\right\|_p.$$

Then, using Lemma 5 and setting $a_i = 1$ for all $i = 1, ..., n$, we have if $0 < \alpha \leq 1$, $\|f(X_1, ..., X_n) - \mathbb{E}f(X_1, ..., X_n)\|_p \leq c_\alpha(\sqrt{p}\left(\sum_{i=1}^{n}\|F_i(X_i, X'_i)\|^2_{\psi_\alpha}\right)^{\frac{1}{2}} + p^{1/\alpha}\max_{1 \leq i \leq n}\|F_i(X_i, X'_i)\|_{\psi_\alpha})$; while if $\alpha > 1$, $\|f(X_1, ..., X_n) - \mathbb{E}f(X_1, ..., X_n)\|_p \leq c'_\alpha(\sqrt{p}(\sum_{i=1}^{n}\|F_i(X_i, X'_i)\|^2_{\psi_\alpha})^{\frac{1}{2}} + p^{1/\alpha}\|(\|F_i(X_i, X'_i)\|_{\psi_\alpha})\|_{\alpha^*})$.

For any $t > 0$, by Markov's inequality,

$$\mathbb{P}\left(|f(X_1, ..., X_n) - \mathbb{E}f(X_1, ..., X_n)| \geq t\right) \leq \frac{\mathbb{E}|f(X_1, ..., X_n) - \mathbb{E}f(X_1, ..., X_n)|^p}{t^p}.$$

By setting $t$ such that $\exp(-p) = \mathbb{E}|f(X_1, ..., X_n) - \mathbb{E}f(X_1, ..., X_n)|^p/t^p$, we get

$$\mathbb{P}\left(|f(X_1, ..., X_n) - \mathbb{E}f(X_1, ..., X_n)| \geq e\|f(X_1, ..., X_n) - \mathbb{E}f(X_1, ..., X_n)\|_p\right) \leq \exp(-p).$$

Let $\delta = \exp(-p)$, we have $p = \log(1/\delta)$ and $0 < \delta < 1/e^2$. Putting the above results together, the proof is complete. $\qquad\square$

## 4 CONCLUSIONS

In this paper, we provided generalization bounds for algorithmic stability with unbounded losses. The technical contribution is a concentration inequality for subweibull random variables. In future work, it would be important to show that some other common learning algorithms, such as stochastic gradient descent, are also stable in the notion of totally Lipschitz stability.

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
