## A    PROOF OF LEMMA 1

*Proof.* Given any samples $S = \{z_1, ..., z_n\} \in \mathcal{Z}^n$ and $S^i = \{z_1, ..., z_{i-1}, z_i', z_{i+1}, ..., z_n\} \in \mathcal{Z}^n$, according to Lemma 7 in (Bousquet & Elisseeff, 2002), for all $i \in [n]$,

$$\mathbb{E}[R(\mathcal{A}, S) - R_n(\mathcal{A}, S)] = \mathbb{E}_{S, z_i'}[\ell(\mathcal{A}_S, z_i') - \ell(\mathcal{A}_{S^i}, z_i')].$$

For fixed $i \in [n]$ and $Z_i^{i-1}$, $Z_{i+1}^n$, define

$$V_i(Z_i, Z_i') = \ell(\mathcal{A}_{Z_1^n}, Z_i') - \ell(\mathcal{A}_{Z_1^{i-1}, Z_i', Z_{i+1}^n}, Z_i').$$

The totally Lipschitz stable condition implies that

$$|V_i(Z_i, Z_i')| \leq \gamma d(Z_i, Z_i').$$

This gives

$$\mathbb{E}[R(\mathcal{A}, S) - R_n(\mathcal{A}, S)] \leq \gamma \mathbb{E} d(Z_i, Z_i'). \tag{1}$$

We now consider two cases separately. In the case of $\alpha > 1$, (1) gives

$$\exp\left(\left(\frac{\mathbb{E}[R(\mathcal{A}, S) - R_n(\mathcal{A}, S)]}{\gamma \|d(Z_i, Z_i')\|_{\psi_\alpha}}\right)^\alpha\right) \leq \exp\left(\left(\frac{|\mathbb{E}[R(\mathcal{A}, S) - R_n(\mathcal{A}, S)]|}{\gamma \|d(Z_i, Z_i')\|_{\psi_\alpha}}\right)^\alpha\right)$$

$$\leq \exp\left(\left(\frac{|\gamma \mathbb{E} d(Z_i, Z_i')|}{\gamma \|d(Z_i, Z_i')\|_{\psi_\alpha}}\right)^\alpha\right) \leq \mathbb{E} \exp\left(\left(\frac{\gamma |d(Z_i, Z_i')|}{\gamma \|d(Z_i, Z_i')\|_{\psi_\alpha}}\right)^\alpha\right) \leq 2,$$

where the third inequality follows from the Jensen's inequality and the last inequality uses the definition $\mathbb{E} \exp\left(\left(\frac{|d(Z_i, Z_i')|}{\|d(Z_i, Z_i')\|_{\psi_\alpha}}\right)^\alpha\right) \leq 2$. Thus, taking logarithms yields the estimate

$$\mathbb{E}[R(\mathcal{A}, S) - R_n(\mathcal{A}, S)] \leq (\log 2)^{1/\alpha} \gamma \|d(Z_i, Z_i')\|_{\psi_\alpha} = (\log 2)^{1/\alpha} \gamma \Delta_\alpha(\mathcal{Z}).$$

In the case of $0 < \alpha \leq 1$,

$$\mathbb{E}[d(Z_i, Z_i')] \leq \int_0^\infty \mathbb{P}(|d(Z_i, Z_i')| > x) dx = \int_0^\infty \mathbb{P}\left(e^{\left(\frac{|d(Z_i, Z_i')|}{\|d(Z_i, Z_i')\|_{\psi_\alpha}}\right)^\alpha} > e^{\left(\frac{x}{\|d(Z_i, Z_i')\|_{\psi_\alpha}}\right)^\alpha}\right) dx$$

$$\leq \int_0^\infty \frac{\mathbb{E}[e^{\left(\frac{|d(Z_i, Z_i')|}{\|d(Z_i, Z_i')\|_{\psi_\alpha}}\right)^\alpha}]}{e^{\left(\frac{x}{\|d(Z_i, Z_i')\|_{\psi_\alpha}}\right)^\alpha}} dx \leq 2 \int_0^\infty e^{-\left(\frac{x}{\|d(Z_i, Z_i')\|_{\psi_\alpha}}\right)^\alpha} dx$$

$$= 2\|d(Z_i, Z_i')\|_{\psi_\alpha} \frac{1}{\alpha} \int_0^\infty e^{-u} u^{\frac{1}{\alpha} - 1} du = 2\|d(Z_i, Z_i')\|_{\psi_\alpha} \frac{1}{\alpha} \Gamma(\frac{1}{\alpha}) = 2\|d(Z_i, Z_i')\|_{\psi_\alpha} \Gamma(\frac{1}{\alpha} + 1).$$

Thus, we get

$$\mathbb{E}[R(\mathcal{A}, S) - R_n(\mathcal{A}, S)] \leq 2\Gamma(\frac{1}{\alpha} + 1)\gamma \|d(X_i, X_i')\|_{\psi_\alpha} = 2\Gamma(\frac{1}{\alpha} + 1)\gamma \Delta_\alpha(\mathcal{Z}).$$

The proof is complete. $\qquad\square$

## B    AUXILIARY LEMMAS

**Lemma 6** (Theorem 1.3.1 in (De la Pena & Giné, 2012)). *Let $a_1, ..., a_n$ a finite non-random sequence, $\{\epsilon_i\}_{i=1}^n$ be a sequence of independent Rademacher variables and $1 < p < q < \infty$. Then,*

$$\|\sum_{i=1}^n \epsilon_i a_i\|_q \leq \left(\frac{q-1}{p-1}\right)^{1/2} \|\sum_{i=1}^n \epsilon_i a_i\|_p.$$

**Lemma 7** (Theorem 2 of (Latała, 1997)). *Let $X_1, ..., X_n$ be a sequence of independent symmetric random variables, and $p \geq 2$. Then,*

$$\frac{e-1}{2e^2} \|(X_i)\|_p \leq \|X_1 + ... + X_n\| \leq e\|(X_i)\|_p,$$

*where $\|(X_i)\|_p := \inf\{t > 0 : \sum_{i=1}^n \log \psi_p(X_i/t) \leq p\}$ with $\psi_p(X) := \mathbb{E}|1 + X|^p$.*

**Lemma 8** (Example 3.2 and 3.3 of (Latała, 1997)). *Assume $X$ be a symmetric random variable satisfying $\mathbb{P}(|X| \geq t) = e^{-N(t)}$. For any $t \geq 0$, we have*

*(a) If $N(t)$ is concave, then $\log \psi_p(e^{-2}tX) \leq pM_{p,X}(t) := \max\{(t^p\|X\|_p^p), (pt^2\|X\|_2^2)\}$.*

*(b) For convex $N(t)$, denote the convex conjugate function $N^*(t) := \sup_{s>0}\{ts - N(s)\}$ and*

$$
M_{p,X}(t) := \begin{cases} p^{-1}N^*(p|t|), & \text{if } p|t| \geq 2 \\ pt^2, & \text{if } p|t| < 2. \end{cases}
$$

*Then $\log \psi_p(tX/4) \leq pM_{p,X}(t)$.*

## C  PROOFS OF LEMMA 3 AND LEMMA 4

*Proof of Lemma 3.* It is enough to prove the monotonicity of function $f(t) = \mathbb{E}h(a+t\epsilon_1)$, for every choice of the parameter $a$. By the convexity assumption we have for $0 < s < t$

$$
\frac{h(a+t) - h(a+s)}{t-s} \geq \frac{h(a-s) - h(a-t)}{t-s}.
$$

Equivalently,

$$
f(s) = \frac{1}{2}(h(a+s) + h(a-s)) \leq \frac{1}{2}(h(a+t) + h(a-t)) = f(t).
$$

The proof is complete. $\qquad\square$

*Proof of Lemma 4.* We will use induction with respect to $n$. For $n = 0$ the statement is obvious, since $\mathbb{E}h(S - \mathbb{E}S) = \mathbb{E}h\left(\sum_{i=1}^{n} \epsilon F_i(X_i, X_i')\right) = h(0)$. Let us thus assume that the Theorem is true for $n-1$. Then

$$
\begin{aligned}
\mathbb{E}h(S - \mathbb{E}S) &= \mathbb{E}h(S - \mathbb{E}_{X_n'}S_n + \mathbb{E}_{X_n}S - \mathbb{E}S) \\
&\leq \mathbb{E}h(S - S_n + \mathbb{E}_{X_n}S - \mathbb{E}S) = \mathbb{E}h(S_n - S + \mathbb{E}_{X_n}S - \mathbb{E}S) \\
&= \mathbb{E}h(\epsilon_n|S - S_n| + \mathbb{E}_{X_n}S - \mathbb{E}S) \\
&\leq \mathbb{E}h(\epsilon_n F_n(X_n, X_n') + \mathbb{E}_{X_n}S - \mathbb{E}S),
\end{aligned}
$$

where the equalities follow from the symmetry, the first inequality follows from the Jensen's inequality and the convexity of $h$, and the last inequality follows from Lemma 3. Now, denoting $Z = \mathbb{E}_{X_n}S$, $Z_i = \mathbb{E}_{X_n}S_i$, we have for $i = 1, ..., n-1$

$$
|Z - Z_i| = |\mathbb{E}_{X_n}S - \mathbb{E}_{X_n}S_i| \leq \mathbb{E}_{X_n}|S - S_i| \leq F_i(X_i, X_i'),
$$

and thus for fixed $X_n$, $X_n'$ and $\epsilon_n$, we can apply the induction assumption to the function $t \rightarrow h(\epsilon_n F_n(X_n, X_n') + t)$ instead of $h$ and $\mathbb{E}_{X_n}S$ instead of $S$, to obtain

$$
\mathbb{E}h(S - \mathbb{E}S) \leq \mathbb{E}h\left(\sum_{i=1}^{n} \epsilon_i F_i(X_i, X_i')\right).
$$

The proof is complete. $\qquad\square$

## D  PROOF OF REMARK 1

*Proof of Remark 1.* The Stirling formula gives a concise form of the term $\sup_{p\geq 2} p^{\frac{-1}{\alpha}}\Gamma^{1/p}(\frac{p}{\alpha} + 1)$. By the Stirling formula

$$
n! = \sqrt{2\pi n}n^n e^{-n+\theta_n}, \quad |\theta_n| < \frac{1}{12n}, n > 1,
$$

we get the following result

$$p^{-\frac{1}{\alpha}}\Gamma^{1/p}(\frac{p}{\alpha}+1) \leq p^{-\frac{1}{\alpha}}(\sqrt{2\pi p/\alpha}(\frac{p}{e\alpha})^{\frac{p}{\alpha}}e^{\frac{\alpha}{12p}})^{1/p}$$

$$= p^{-\frac{1}{\alpha}}(\sqrt{\frac{2\pi}{\alpha}})^{1/p}p^{1/2p}(\frac{p}{\alpha})^{1/\alpha}e^{\alpha/12p^2-1/\alpha}$$

$$\leq (\sqrt{\frac{2\pi}{\alpha}})^{1/p}e^{1/2e}\frac{1}{(e\alpha)^{1/\alpha}}e^{\alpha/12p^2}$$

$$\leq (\sqrt{\frac{2\pi}{\alpha}})^{1/2}e^{1/2e}\frac{1}{(e\alpha)^{1/\alpha}}e^{\alpha/48},$$

where the first inequality uses the Stirling formula and the second inequality uses the fact that $p^{1/p} \leq e^{1/e}$. $\qquad\square$