# OpenReview forum: "Algorithmic Stability Unleashed: Generalization Bounds with Unbounded Losses"
_ICLR.cc/2024/Conference — Submitted to ICLR 2024_

### Official Review · Reviewer_wVug · 2023-10-24

**Soundness:** 2 fair
**Presentation:** 3 good
**Contribution:** 2 fair
**Rating:** 5
**Confidence:** 4

**Summary:**

The paper studies high-probability generalization bounds for stable algorithms with unbounded loss functions. In particular, the paper considers a symmetric and totally Lipschitz stable algorithm, where the change of the output model with respect to the perturbation of a single example follows from a subweibull distribution. To this aim, the paper first builds a bound on the p-norm of a summation of independent subweibull random variables, and apply it to develop concentration inequalities for random functions with the increment behaving as a subweibull random variable.

**Strengths:**

The paper studies learning problems with unbounded loss functions. The assumption on subweibull distribution is more relaxed than the previous sub-gaussian and sub-exponential distribution. The derived results recover the previous results for both the sub-gaussian and sub-exponential case.

**Weaknesses:**

The high-probability bound established in the paper is of the order $O(\sqrt{n}\gamma)$, which can only imply sub-optimal bounds. One would be more interested in bounds of the order $O(1/\sqrt{n}+\gamma\log n)$, which were developed for learning with bounded loss functions.

The analysis seems to be not clear and there are several issues in the theoretical analysis. Furthermore, the main result is based on Lemma 5, which seems to be a corollary of the results in Latała, 1997. The analysis in Latała, 1997 gives general bounds on the p-norm of a summation of random variables. Lemma 5 is derived by considering the special subweibull random variables.

The contribution seems to be a bit incremental. The problem setting follows the concentration in unbounded metric spaces established in Kontorovich (2014). The main contributions are the extension of the analysis in Kontorovich (2014), Maurer & Pontil (2021) from sub-gaussian and sub-exponential distribution to subweibull distribution. It is not clear to me whether the extension is substantial, and the paper does not include enough examples to justify the importance of subweibull distribution in practice.

**Questions:**

- In the proof of Lemma 5, the paper shows that $\|\sum_i a_iX_i\| \leq \|\sum_i a_i Z_i\|$. It is not clear to me how this inequality holds. Indeed, the random variable $Z_i$ is not clearly defined, and the paper only assumes $P(|Z_i|\geq t)=\exp(-t^\alpha)$. However, it is not clear to me how to use this property to derive $\|\sum_ia_iX_i\|\leq \|\sum_ia_iZ_i\|$.

- In the proof of Lemma 5, the paper uses the inequality $\inf\{t>0:\sum_i\max\{X_i(t),Y_i(t)\}\leq p\}\leq \inf\{t>0:\sum_iX_i(t)\leq p\}+\inf\{t>0:\sum_iY_i(t)\leq p\}$. It is not clear to me how this inequality holds. Would you please provide details? Furthermore, you have $p(a_ie^2/t)^2\|Z_i\|^2\leq p$ in the second line, but the last line is $p^2\Gamma(2/\alpha+1)e^4/t^2\|a\|^2 \leq p$.

- In the proof of Lemma 5, you have the identity $p^{-1-(p-2)/\alpha}=p^{-p/\alpha}p^{(2-\alpha)/\alpha}$, which is not correct.

- The generalization bounds have very complicated dependency on the parameter $\alpha$, which makes the results hard to interpret. How should we understand the effect of $\alpha$ on generalization?

---

> ### Author Response · Authors · 2023-11-22
> **Part I**
>
> Thanks for your time and thoughts. We will answer all your questions.
>
> Question 1: Issue in Weakness ``The high-probability bound established in the paper is ...''
>
> Answer: Recently, (Bousquet et al. 2020) provided a sharper generalization bound of the order $O(1/\sqrt{n} + \gamma \log n)$ for algorithmic stability. In their proof, they also use the $p$-th moment technique, and the $p$-th moment version of the Bounded differences/McDiarmid’s inequality is an important component. It seems to be straightforward to give better generalization inequality by combining our $p$-th moment inequality and the technique of peeling the dataset in (Bousquet et al. 2020). Since our work mainly follows the work  (Kontorovich, 2014; Maurer \& Pontil, 2021), we didn't consider giving sharper generalizations in the manuscript.
>  We will include this part of the results in the final version.
>
> Question 2: Issue in Weakness ``The analysis seems to be not clear and there are several issues ... ''
>
> Answer: Our work is not simply a corollary of the results in Latała, 1997. The analysis in (Latała, 1997) gives bounds for the \emph{symmetric} random variables, which can not be directly used in our analysis.
> On one hand, we need to carefully construct new random variables to satisfy the symmetry condition, for example, we introduce random variables $Y_i$, $Z_i$ in the proof of Lemma 5. On the other hand, since we study the weighted summation, Khinchin-Kahane inequality is also required to use.
>
> Question 3: Issue in Weakness ``The contribution seems to be a bit incremental. The problem setting follows ... ''
>
> Answer: We believe our work is not incremental. Here, we discuss the technical novelties of Theorem 1. The first challenge to prove Theorem 1 is that we deal with the concentration of the general function $f$. Actually, the related work (Kontorovich, 2014) used the martingale method to decompose $f-\mathbb{E}f$, while (Maurer and Pontil, 2021) use the sub-additivity of entropy to decompose the general function $f-\mathbb{E}f$. After the decomposition, the next step of (Kontorovich, 2014) and (Maurer and Pontil, 2021) is to bound the MGF ($\mathbb{E}e^{\lambda Z}$) or a variant MGF ($\mathbb{E}Z^2 e^{\lambda Z}$), respectively. The second challenge is that the MGF is bounded for sub-Gaussian and sub-exponential random variables, but it is unbounded for subWeibull variables because there is some convexity lost. The standard technique to prove the MGF failed for the heavy-tailed subWeibull random variables, as discussed in the penultimate paragraph of the Introduction and Remark 1. This implies that if we do not study the MGF, we need to consider different decomposition on the general function $f-\mathbb{E}f$.
>
> To address the first challenge, we introduce Lemma 4, where we decompose the general function $f-\mathbb{E}f$ to the sum of independent sub-Weibul variables. In the proof of Lemma 4, a key step is that we need to construct a function $t \to h(\epsilon_n F_n(X_n, X'_n) +t)$ to apply our induction assumption. The proof of Lemma 4 may be simple, but Lemma 4 itself is very useful. For example, one can use Lemma 4 to prove more McDiarmid inequalities, e.g., the polynomially decaying random variables, which will enrich the family of McDiarmid inequality.
>
> To address the second challenge, rather than bounding the MGF, we bound the $p$-th moment of the sum of subWeibull random variables. Thanks to the fact that subWeibull random variables are log-convex for $\alpha \leq 1$ and log-concave for $\alpha \geq 1$, we can apply Lata{\l}a's inequality (Lemma 7 and Lemma 8) to bound this $p$-th moment. However, it is not a direct application of the Lata{\l}a's inequality. Lata{\l}a's inequality holds for the $p$-th moment of the \emph{symmetric} random variables. On one hand, we need to carefully construct new random variables to satisfy the symmetry condition, for example, we introduce random variables $Y_i$, $Z_i$ in the proof of Lemma 5. On the other hand, since we study the weighted summation, Khinchin-Kahane inequality is also required to use.
>
> Finally, we would like to quote the comment of Reviewer LoHq to confirm the technical novelties: "The proof consists of two steps: reduction to sum of independent sub-Weibul variables (Lemma 4), and the corresponding moment bound on a sum of sub-weibull rv's. The first part is done thanks to a conditioning trick with Jensen's inequality. The second part is done thanks to a carful application of Latala's inequality (Lemma 7 in the appendix)."
>
> Proving generalization bounds for heavy-tailed losses is a significant direction in the learning theory community. For the tool of PAC-Bayes, Space Complexity, and Information theory, there are many works that study the heavy-tailed losses, while it is lacking for the tool of stability. Our work is the first attempt to give generalization bounds for the heavy-tailed loss for Algorithmic Stability. Therefore, we believe our extension is substantial.

---

> ### Author Response · Authors · 2023-11-22
> **Part II**
>
> Question 4: Issue in Question 1 ``In the proof of Lemma 5, the paper shows ...''
>
> Answer: Thanks for this review. Let $\\{Z_i\\}^n_{i=1}$ be independent symmetric random variables satisfying $|Z_i| \overset{d}{=} Y_i$ and $\mathbb{P}(|Z_i| \ge t) = \exp(-t^\alpha)$ for all $t \ge 0$. The symmetric variable $Z_i$ is introduced to use the Lata{\l}a's inequality.
>
>  $ || \sum\_{i=1}^p  a\_i X\_i ||\_p \leq || \sum_{i=1}^p a\_i Z\_i ||\_p $? Did the reviewer mean $|| \sum\_{i=1}^p a\_i  \epsilon\_i Y\_i ||\_p \leq || \sum\_{i=1}^p a\_i Z\_i ||\_p $?
> Rigorously, it should be $|| \sum\_{i=1}^p a\_i  \epsilon\_i Y\_i ||\_p = || \sum_{i=1}^p a\_i Z\_i ||\_p $. Note that the tail of $Y\_i$ is $ P(Y\_i \geq t) \leq e^{-t^{\alpha}}$, the $Y\_i$ are positive since $Y\_i = (|X\_i| - (\log 2)^{1/\alpha})\_+$, and tail of  $Z_i$ is also $\mathbb{P}(|Z\_i| \ge t) = \exp(-t^\alpha)$.  According to the equivalence of the tail and the moment, it is clear that  $ || \sum\_{i-1}^p a\_i \epsilon\_i Y\_i ||\_p = || \sum\_{i=1}^p a\_i Z\_i ||\_p $.
>
>
> Question 5: Issue in Question 2 ``In the proof of Lemma 5, the paper uses the inequality ...''
>
> Answer:  Thanks for this review.
>
> (1) Sorry that we missed a constant $2$ in this step. The inequality is that
>
> \begin{align*}
> \inf [ t>0 : \sum\_i \max [ X\_i(t), Y\_i(t) ]  \leq p ] \leq \inf [ t>0 : \sum\_i X\_i(t) +
>  \sum\_i Y\_i(t)  \leq p ]
>  \leq \inf [ t>0 : 2 \sum\_i X\_i(t)  \leq p ] + \inf [ t>0: 2 \sum\_i X\_i(t)  \leq p ].
> \end{align*}
>
>  We have fixed this issue.
>
> (2) For the next question, the inequality holds because $||Z_i||\_p^p = p \Gamma(\frac{p}{\alpha} + 1)$ and we set $p=2$.
>
>
> Question 6: Issue in Question 3 ``In the proof of Lemma 5, you have the identity ...''
>
> Answer:
> $p^{-1-(p-2)/\alpha} = p^{-1-\frac{(p-2)}{\alpha}}  = p^{-p/\alpha}p^{(2-\alpha)/\alpha}$.
> This equality is correct, please check it again.
>
> Question 7: Issue in Question 4 ``The generalization bounds have very complicated dependency ...''
>
> Answer:
> (1) We first discuss Theorem 1.
> $c_\alpha =2 \sqrt{2} ((\log 2)^{1/\alpha}+e^3\Gamma^{1/2}(\frac{2}{\alpha} +1) + e^3 3^{\frac{2-\alpha}{3\alpha}} \sup_{p\ge 2}p^{\frac{-1}{\alpha}} \Gamma^{1/p}(\frac{p}{\alpha} +1) )$. According to the property of the Gamma function, $\Gamma(\frac{2}{\alpha} +1)$ becomes bigger as $\alpha$ becomes smaller. As for the the term $\sup_{p \geq 2} p^{\frac{-1}{\alpha}} \Gamma^{1/p}(\frac{p}{\alpha} +1) )$,  the Stirling formula easily gives a concise form.
> By the Stirling formula
> \begin{align*}
> n! = \sqrt{2\pi n}n^ne^{-n + \theta_n}, \quad |\theta_n|<\frac{1}{12 n}, n>1,
> \end{align*}
> we get the following result
> \begin{align*}
>     p^{-\frac{1}{\alpha}} \Gamma^{1/p}(\frac{p}{\alpha} + 1)
>     \leq
>     p^{-\frac{1}{\alpha}} (\sqrt{2\pi p/\alpha} (\frac{p}{e\alpha})^{\frac{p}{\alpha}} e^{\frac{\alpha}{12p}} )^{1/p}
>     = p^{-\frac{1}{\alpha}} (\sqrt{\frac{2\pi}{\alpha}})^{1/p} p^{1/2p} (\frac{p}{\alpha})^{1/\alpha} e^{\alpha/12p^2 -1/\alpha}
>     \leq  (\sqrt{\frac{2\pi}{\alpha}})^{1/p}  e^{1/2e} \frac{1}{(e\alpha)^{1/\alpha}} e^{\alpha/12p^2}
>     \leq  (\sqrt{\frac{2\pi}{\alpha}})^{1/2}  e^{1/2e} \frac{1}{(e\alpha)^{1/\alpha}} e^{\alpha/48},
> \end{align*}
> where the first inequality uses the Stirling formula and the second inequality uses the fact that $p^{1/p} \leq e^{1/e}$. Thus we can find that the term $\sup_{p \geq 2}p^{\frac{-1}{\alpha}} \Gamma^{1/p}(\frac{p}{\alpha} +1) )$ is not involved in $p$ and only depends on $\alpha$, and the smaller $\alpha$ is, the bigger this term is.
> According to above analysis, we  have the conclusion that the smaller $\alpha$ is, the constant related to $\alpha$ (i.e., $c_\alpha$) in the upper bound is bigger.  We then discuss Theorem 2. As discussed in Section 2.1, the smaller $\alpha$ is, the heavier tail the random variable has, which means that $\Delta_\alpha$ is bigger according to the definition. By the above analysis, we  have the conclusion that the smaller $\alpha$ is, the terms related to $\alpha$ ($c_\alpha$ and $\Delta_\alpha$) in the upper bound is bigger. This result is consistent with the intuition: heavier-tailed distribution, i.e., smaller $\alpha$, will lead to a larger generalization bound.
>
> References
>
> Olivier Bousquet and Andre Elisseeff. Stability and Generalization. 2002.
>
> Aryeh Kontorovich. Concentration in unbounded metric spaces and algorithmic stability. 2014.
>
> Vitaly Feldman and Jan Vondrak. High probability generalization bounds for uniformly stable algorithms with nearly optimal rate. 2019.
>
> Oliver Bousquet, Yegor Klochkov, and Nikita Zhivotovskiy. Sharper bounds for uniformly stable algorithms 2020.
>
> Andreas Maurer and Massimiliano Pontil. Concentration inequalities under sub-gaussian and sub-exponential conditions. 2021

---

### Official Review · Reviewer_LoHq · 2023-10-29

**Soundness:** 3 good
**Presentation:** 4 excellent
**Contribution:** 2 fair
**Rating:** 6
**Confidence:** 5

**Summary:**

The present paper proposes a stability generalization bound for a variant of non-uniform stability condition.

Their analysis revolves around a new form of McDiarmid inequality, where the differences are bounded by another function, which satisfies certain moment conditions (sub-Weibul). The proof consists of two steps: reduction to sum of independent sub-Weibul variables (Lemma 4), and the corresponding moment bound on a sum of sub-weibull rv's. The first part is done thanks to a conditioning trick with Jensen's inequality. The second part is done thanks to a carful application of Latala's inequality (Lemma 7 in the appendix). This inequality is a direct extension of one from Kontorovich (2014) to an interesting non-trivial case. I particularly like that Theorem 1 does not have any extra logarithms.

It would be great to include some analysis of how optimal is the bound. Whether there are some cases, where each of the terms is necessary. For example, is it possible to replace the first term with \sqrt{\sum_{i = 1} E (F_i)^{2}}, i.e. without the orlizc norm? I understand how to do it if we can sacrifice additional logorithm in the second term (e.g. using Thm 15.10 in [1]), is it possible to do it without this sacrifice?

My other concern is about sub optimality of the generalization bounds listed on pages 5-6. Since the the original paper Bousquet & Eliseef (2002), there have beed significant improvements in these bounds [2, 3]. The proof in [3] works directly with moment bounds, perhaps it is possible to make it work with their technique and obtain a better generalization inequality. Since you mention generalization bounds in the title, I think it is crucial to make the bound closer to state-of-the-art.

I also want to point out that the condition $|S - S_i| \leq F_i(X_i, X_i')$ is also to some extent uniform, since it does not depend on $X_1, \dots, X_{i-1}, X_{i+1}, \dots, X_{n}$. Does the condition hold e.g. for ridge regression that you mention in the introduction?

To sum up, although the paper provides interesting results, there is room for immediate improvement.

[1] Boucheron et al (2014) Concentration inequalities.

[2] Feldman and Vondrak (2019). Generalization bounds for uniformly stable algorithms.

[3] Bousquet et al (2020). Sharper bounds for uniformly stable algorithms

**Strengths:**

Novel version of bounded differences inequality, where the differences are not really bounded.

**Weaknesses:**

Do not touch question of optimality.

Suboptimal generalization bound.

**Questions:**

Does the condition hold e.g. for ridge regression that you mention in the introduction?

---

> ### Author Response · Authors · 2023-11-22
>
> Thank you for your review. We will answer all your questions.
>
> Question 1: Issue in Summary ``It would be great to include some analysis of how optimal ...''
>
> Answer:  Yes, the bound provided by Theorem 1 is solely in terms
> of $||F\_i (X\_i, X'\_i)||\_{\psi\_\alpha}$.  It is possible to replace the orlizc norm with the variance $\sqrt{\sum\_{i=1} \mathbb{E}(F\_i)^2} $, as you commented, however, an additional logarithms $\log^{\frac{1}{\alpha}} (n+1)$ appeared in the second term, which is induced by taking the max operator outside the orlicz norm. Our method of using the orlicz norm does not introduce the logarithmic term. But we think this review is very constructive. From the classical central limit theorem (asymptotically
> the distribution of the sum (properly scaled) is determined by the variance of the sum), we naturally expect the Gaussian part of the tail to depend on the variance $\sqrt{\sum_{i=1} \mathbb{E}(F_i)^2} $ only.  We are working to provide a tighter bound without sacrificing the additional logarithm.
>
> We now provide some discussions on the optimality of our bounds. In Theorem 1, if $0<\alpha\leq1$, our inequality
> \begin{align*}
> |f(X\_1,...,X\_n) - \mathbb{E} f(X\_1,...,X\_n)| \le c\_\alpha \left ( \sqrt{\log(\frac{1}{\delta})} \left(\sum\_{i=1}^n ||F\_i (X\_i, X'\_i)||\_{\psi_\alpha}^2\right)^{\frac{1}{2}} + \log^{1/\alpha}(\frac{1}{\delta}) \max\_{1\le i \le n}||F\_i (X\_i, X'\_i)||_{\psi\_\alpha}\right)
> \end{align*}
> shows a mixture of
> sub-gaussian $\sqrt{\log(\frac{1}{\delta})}$
> and sub-weibull $\log^{1/\alpha}(\frac{1}{\delta})$ tails. The sub-Gaussian tail  is of course expected from the central limit theorem, and the sub-weibull tail captures the right decaying rate of the sub-weibull random variables. Therefore, our inequality successfully captures the right sub-Gaussian tail for small deviations and the right sub-weibull tail for large deviations, which also means that the convergence rate will be faster for small deviations and will be slower for large deviations.
>
> If $\alpha >1$, our inequality
> \begin{align*}
> |f(X\_1,...,X\_n) - \mathbb{E} f(X\_1,...,X\_n)|   \le&c'\_\alpha\left (\sqrt{\log(\frac{1}{\delta})} \left(\sum\_{i=1}^n ||F\_i (X\_i, X'\_i)||\_{\psi\_\alpha}^2\right)^{\frac{1}{2}} +  \log^{1/\alpha}(\frac{1}{\delta}) || (|| F (X, X')||\_{\psi_\alpha}) ||\_{\alpha^\ast}\right)
> \end{align*}
> follows the same spirit and also exhibits a mixture of
> sub-gaussian $\sqrt{\log(\frac{1}{\delta})}$
> and sub-weibull $\log^{1/\alpha}(\frac{1}{\delta})$ tails.
>
> Question 2: Issue in Summary ``My other concern is about sub optimality of the generalization bounds ...''
>
> Answer: Recently, (Bousquet et al. 2020) provided a sharper generalization bound for algorithmic stability. In their proof, they also use the $p$-th moment technique, and the $p$-th moment version of the Bounded differences/McDiarmid’s inequality is an important component. It seems to be straightforward to give better generalization inequality by combining our $p$-th moment inequality and the technique of peeling the dataset in (Bousquet et al. 2020). Since our work mainly follows the work  (Kontorovich, 2014; Maurer \& Pontil, 2021), we didn't consider giving sharper generalizations in the manuscript.
>  We will include this part of the results in the final version.
>
> Question 3: Issue in Summary ``I also want to point out that the condition ...''
>
>
> Answer: Yes. This condition holds for ridge regression. Here are the proofs.
>
> Let's consider that the loss function $f(x,y) = (h(x) - y)^2 $, $h: \mathcal{X} \to [0,1]$ is a Lipschitz function with Lipschitz constant $L$: $h(x) - h(y) \leq L d(x,y)$, and $(\mathcal{Z}, d_2)$ is the metric
> space where $\mathcal{Z} = \mathcal{X} \times [0,1]$ and $d_2((x,y), (x',y') ) = (d(x,x')^2 + |y-y'|^2)^{1/2}$. Let us take $\psi[0,1]^2 \to \mathbb{R}$ to be $\psi(h,y) = (h-y)^2$, which satisfies $\max_{(h,y) \in [0,1]^2} \| \nabla \psi(h,y) \|_2 =2^{3/2}$.
> It follows that
> \begin{align}
> |f(x,y) - f(x',y')| & = | (h(x) -y)^2 - (h(x') -y')^2 |
>  \leq 2^{3/2} ((h(x) -h(x'))^2 + (y - y')^2)^{1/2}
>  \leq 2^{3/2} (L^2 d(x ,x')^2 + (y - y')^2)^{1/2}  \leq 2^{3/2} \max [ 1, L ] d_2((x,y), (x',y')).
> \end{align}
> When $h(x) = ax $, we have $|f(x,y) - f(x',y')| \leq 2^{3/2} \max [ 1, a ] d_2((x,y), (x',y'))$. Corresponding $F_i(X_i,X'_i))$ to $d_2((x,y), (x',y'))$, we can conclude that the condition $|S-S_i| \leq F_i(X_i,X'_i)$ holds for ridge regression.
>
> References
>
> Olivier Bousquet and Andre Elisseeff. Stability and Generalization. 2002
>
> Aryeh Kontorovich. Concentration in unbounded metric spaces and algorithmic stability. 2014
>
> Vitaly Feldman and Jan Vondrak. High probability generalization bounds for uniformly stable algorithms with nearly optimal rate. 2019
>
> Oliver Bousquet, Yegor Klochkov, and Nikita Zhivotovskiy. Sharper bounds for uniformly stable algorithms 2020
>
> Andreas Maurer and Massimiliano Pontil. Concentration inequalities under sub-gaussian and sub-exponential conditions. 2021

---

### Official Review · Reviewer_K21p · 2023-11-01

**Soundness:** 2 fair
**Presentation:** 1 poor
**Contribution:** 2 fair
**Rating:** 3
**Confidence:** 3

**Summary:**

The paper studies algorithmic stability and generalization bounds for unbounded losses of independent subweibull random variables. Specifically, the concentration inequality for subweibull random variables (which covers subexponential or subgaussian random variables studied in the previous works) is established for general losses for both heavy tails and non-heavy tails variables. Based on this result, the generalization bounds in the order of $O(1/\sqrt{n})$ for the symmetric and Lipschitz stable algorithms are derived, which recovers the previous results for subexponential or subgaussian distributions up to some constants.

**Strengths:**

A more general stability and generalization results for the symmetric and Lipschitz stable algorithms are established by introducing the subweibull diameter of the input space. The concentration inequalities (Theorem 1) might be of independent interest.

**Weaknesses:**

1. Although the paper extends the previous results on subexponential or subgaussian random variables to a more general setting, it seems that it is incremental. There is no technological or conceptual novelty here. The key result of the paper is Theorem 1, while the proofs are standard and straightforward. The generalization bounds can be easily obtained once the connection between the stability and generalization is established.

2. The paper is worse written. Discussions of the main results (Theorems 1 and 2) are insufficient. For example, the bounds in Theorem 1 are related to $c_{\alpha}$, which are dependent on $\Gamma^{1/2}(\alpha)$ and a $sup_{p\ge 2}$ term.  The impact of the value of $\alpha$ on the results needs to be discussed since it controls the degree of heavy tails of the random variable. Similarly, the value of $\Delta_{\alpha}$ should be discussed in Theorem 2. In addition, I would suggest the authors go through the paper and polish the language, some notations are not introduced and some symbols are inconsistent. For example, Page 4, the learning algorithm is sometimes notated by $A_S$ and sometimes $\mathcal{A}_S$.

**Questions:**

1.	As mentioned in weakness 1, it seems that the proofs of the theorems are standard. Could the authors point out the technique novelty of the paper?
2.	The paper establishes the generalization bounds for the Lipschitz losses. It is possible to remove this assumption, or could we assume that the loss is smooth instead?
3.	What does “\mathcal{A} is a symmetric algorithm” mean in Lemma 1?

---

> ### Author Response · Authors · 2023-11-22
> **Part I**
>
> Thank you for your review. Here are the replies for your questions.
>
> Question 1: As mentioned in weakness 1, it seems that the proofs ...
>
> Answer: We believe our work is not incremental. Here, we discuss the technical novelties of Theorem 1. The first challenge to prove Theorem 1 is that we deal with the concentration of the general function $f$. Actually, the related work (Kontorovich, 2014) used the martingale method to decompose $f-\mathbb{E}f$, while (Maurer and Pontil, 2021) use the sub-additivity of entropy to decompose the general function $f-\mathbb{E}f$. After the decomposition, the next step of (Kontorovich, 2014) and (Maurer and Pontil, 2021) is to bound the MGF ($\mathbb{E}e^{\lambda Z}$) or a variant MGF ($\mathbb{E}Z^2 e^{\lambda Z}$), respectively. The second challenge is that the MGF is bounded for sub-Gaussian and sub-exponential random variables, but it is unbounded for subWeibull variables because there is some convexity lost. The standard technique to prove the MGF failed for the heavy-tailed subWeibull random variables, as discussed in the penultimate paragraph of the Introduction and Remark 1. This implies that if we do not study the MGF, we need to consider different decomposition on the general function $f-\mathbb{E}f$.
>
> To address the first challenge, we introduce Lemma 4, where we decompose the general function $f-\mathbb{E}f$ to the sum of independent sub-Weibul variables. In the proof of Lemma 4, a key step is that we need to construct a function $t \to h(\epsilon_n F_n(X_n, X'_n) +t)$ to apply our induction assumption. The proof of Lemma 4 may be simple, but Lemma 4 itself is very useful. For example, one can use Lemma 4 to prove more McDiarmid inequalities, e.g., the polynomially decaying random variables, which will enrich the family of McDiarmid inequality.
>
> To address the second challenge, rather than bounding the MGF, we bound the $p$-th moment of the sum of subWeibull random variables. Thanks to the fact that subWeibull random variables are log-convex for $\alpha \leq 1$ and log-concave for $\alpha \geq 1$, we can apply Lata{\l}a's inequality (Lemma 7 and Lemma 8) to bound this $p$-th moment. However, it is not a direct application of the Lata{\l}a's inequality. Lata{\l}a's inequality holds for the $p$-th moment of the \emph{symmetric} random variables. On one hand, we need to carefully construct new random variables to satisfy the symmetry condition, for example, we introduce random variables $Y_i$, $Z_i$ in the proof of Lemma 5. On the other hand, since we study the weighted summation, Khinchin-Kahane inequality is also required to use.
>
> Finally, we would like to quote the comment of Reviewer LoHq to confirm the technical novelties: "The proof consists of two steps: reduction to sum of independent sub-Weibul variables (Lemma 4), and the corresponding moment bound on a sum of sub-weibull rv's. The first part is done thanks to a conditioning trick with Jensen's inequality. The second part is done thanks to a carful application of Latala's inequality (Lemma 7 in the appendix)."
>
> Establishing the connection between the stability and generalization needs to consider different cases: $\alpha>1$ and $0<\alpha\le 1$. Due to the fact that there is some convexity lost in the case $0<\alpha\le 1$, the proof excludes the Jensen’s inequality and requires a different analysis.
>
> Question 2: The paper establishes the generalization bounds for the Lipschitz losses ...
>
> Answer:
> Our generalization bound is derived by the established McDiarmid inequality, and a Lipschitz-type assumption is typically required for the McDiarmid inequality. Thus, this paper establishes the generalization bounds for the Lipschitz losses. But we think this review is inspiring.
> The smoothness itself is a Lipschitz-type assumption on the gradient. When assuming that the loss is smooth instead, it is possible to establish generalization bounds on the gradient by our concentration inequality. The generalization bound on the gradient is drawing increasing interest in the nonconvex learning of the learning theory community.
>
>
>
>
> Question 3: What does ``$\mathcal{A}$ is a symmetric algorithm'' mean in Lemma 1?
>
> Answer: This means that the algorithm's outputs remain unchanged when swapping the order of samples in the data set.

---

> ### Author Response · Authors · 2023-11-22
> **Part II**
>
> Question 4: Issue in Weakness 2. ``The paper is worse written. Discussions of the main results ...''
>
> Answer:
>
> (1) We first discuss Theorem 1.
> $c_\alpha =2 \sqrt{2} ((\log 2)^{1/\alpha}+e^3\Gamma^{1/2}(\frac{2}{\alpha} +1) + e^3 3^{\frac{2-\alpha}{3\alpha}} \sup_{p\ge 2}p^{\frac{-1}{\alpha}} \Gamma^{1/p}(\frac{p}{\alpha} +1) )$. According to the property of the Gamma function, $\Gamma(\frac{2}{\alpha} +1)$ becomes bigger as $\alpha$ becomes smaller. As for the the term $\sup_{p \geq 2}$,  the Stirling formula easily gives a concise form.
> By the Stirling formula
> \begin{align*}
> n! = \sqrt{2\pi n}n^ne^{-n + \theta_n}, \quad |\theta_n|<\frac{1}{12 n}, n>1,
> \end{align*}
> we get the following result
> \begin{align*}
>     p^{-\frac{1}{\alpha}} \Gamma^{1/p}(\frac{p}{\alpha} + 1)
>     \leq
>     p^{-\frac{1}{\alpha}} (\sqrt{2\pi p/\alpha} (\frac{p}{e\alpha})^{\frac{p}{\alpha}} e^{\frac{\alpha}{12p}} )^{1/p}
>     &= p^{-\frac{1}{\alpha}} (\sqrt{\frac{2\pi}{\alpha}})^{1/p} p^{1/2p} (\frac{p}{\alpha})^{1/\alpha} e^{\alpha/12p^2 -1/\alpha}
>     \leq  (\sqrt{\frac{2\pi}{\alpha}})^{1/p}  e^{1/2e} \frac{1}{(e\alpha)^{1/\alpha}} e^{\alpha/12p^2}
>     \leq  (\sqrt{\frac{2\pi}{\alpha}})^{1/2}  e^{1/2e} \frac{1}{(e\alpha)^{1/\alpha}} e^{\alpha/48},
> \end{align*}
> where the first inequality uses the Stirling formula and the second inequality uses the fact that $p^{1/p} \leq e^{1/e}$. Thus we can find that the term $\sup_{p \geq 2}$ is not involved in $p$ and only deoends on $\alpha$, and the smaller $\alpha$ is, the bigger this term is.
> According to above analysis, we  have the conclusion that the smaller $\alpha$ is, the constant related to $\alpha$ (i.e., $c_\alpha$) in the upper bound is bigger.  We then discuss Theorem 2. As discussed in Section 2.1, the smaller $\alpha$ is, the heavier tail the random variable has, which means that $\Delta_\alpha$ is bigger according to the definition. By the above analysis, we  have the conclusion that the smaller $\alpha$ is, the terms related to $\alpha$ ($c_\alpha$ and $\Delta_\alpha$) in the upper bound is bigger. This result is consistent with the intuition: heavier-tailed distribution, i.e., smaller $\alpha$, will lead to a larger upper bound.
>
> (2) Thanks for this review. We have revised $A_S$ to $\mathcal{A}_S$. We will further go through the paper and polish the language in the final version.
>
>
> References
>
> Olivier Bousquet and Andre Elisseeff. Stability and Generalization. 2002.
>
> Aryeh Kontorovich. Concentration in unbounded metric spaces and algorithmic stability. 2014.
>
> Vitaly Feldman and Jan Vondrak. High probability generalization bounds for uniformly stable algorithms with nearly optimal rate. 2019.
>
> Oliver Bousquet, Yegor Klochkov, and Nikita Zhivotovskiy. Sharper bounds for uniformly stable algorithms 2020.
>
> Andreas Maurer and Massimiliano Pontil. Concentration inequalities under sub-gaussian and sub-exponential conditions. 2021

---

### Official Review · Reviewer_wNWA · 2023-11-04

**Soundness:** 3 good
**Presentation:** 2 fair
**Contribution:** 2 fair
**Rating:** 5
**Confidence:** 3

**Summary:**

Algorithmic stability has increasingly become an important tool for studying the generalization ability of machine learning algorithms. The general idea is to understand how well algorithms generalize out-of-sample by looking that how insensitive ('stable") they are to perturbations of the input dataset. Intuitively, algorithms that do not have high dependence on any small subset of the dataset tend to not overfit to the data. This is should be contrasted with uniform convergence where generalization is proven using the simplicity of the class of possible output hypotheses instead. The present paper studies generalization under algorithmic stability for unbounded losses, complementing the standard techniques that works of case of algorithms that satisfy uniform boundedness.

The paper presents results extending uniform boundedness to a notion they call subWeibull diameter which corresponds to control on the tails of the deviations and show algorithmic stability bounds under this condition. The main technical contribution is a concentration inequality for subWeibull distributions that could be of independent interest.

**Strengths:**

The paper studies a natural extension of bounded uniform stability to unbounded losses and notes an interesting concentration inequality for Subweibull distributions. I believe the concentration inequality could be of general use in other applications.

**Weaknesses:**

Though the generalization to unbounded losses is interesting, it would be nice if there are more applications presented justifying the generalization to this case. Also, see author questions section for detailed.

I would be happy to raise my score if these concerns are addressed.

**Questions:**

- One point that would be nice to emphasize the technical novelties in the proof. This is not meant to sound critical but from a first reading it is difficult to fully appreciate the challenge of the proofs As stated below, Lemma 3 and 4 are standard (contraction and symmetrization). So the main lemma seems to be Lemma 5. From my reading it is difficult to figure out why this is substantially different from the standard proof that Orlicz spaces ( for phi = exp(t^p) - 1 for p>1 ) have bounded moments. I understand that there is some convexity lost for p < 1 which is what you study but from the proof as written I fail to see the challenge (other than the fact that one can't go directly through the MGF. For example I am thinking of the proof here (https://www.math.uci.edu/~rvershyn/papers/HDP-book/HDP-book.pdf Prop 2.5.2). The challenge seems to be showing sums of subWeibull satisfy subWeibull tails (or rather keeping track of the parameters; ) without using the MGF (if this interpretation is correct maybe it is useful to emphasize). In summary would be nice to have a part explaining the technical novelty
- I would recommend moving the proofs of Lemma 3 and Lemma 4 to the appendix since they are standard to allow the reader to focus on the new contributions of the work.
- I might be missing something simple but I don't see how one went from a bound in terms of F to a bound in terms of the Weibull diameter without further assumptions on f/F. A priori F is some arbitrary function of F(X_i , X'_i) right, unrelated to the distance? I guess that in the remark there is some Lipshitzness assumed. It would be nice to make that explicit.
- Page 4: Population Risk

---

> ### Author Response · Authors · 2023-11-22
> **Part I**
>
> Thank you for your time and thoughts. Below, we reply to all the issues.
>
> Question 1: One point that would be nice to emphasize the technical novelties in the proof...
>
> Answer: Thanks for this review. The first challenge is that we deal with the concentration of the general function $f$. Actually, the related work (Kontorovich, 2014) used the martingale method to decompose $f-\mathbb{E}f$, while (Maurer and Pontil, 2021) use the sub-additivity of entropy to decompose the general function $f-\mathbb{E}f$. After the decomposition, the next step of (Kontorovich, 2014) and (Maurer and Pontil, 2021) is to bound the MGF ($\mathbb{E}e^{\lambda Z}$) or a variant MGF ($\mathbb{E}Z^2 e^{\lambda Z}$), respectively. The second challenge is that, as you commented, the MGF is bounded for sub-Gaussian and sub-exponential random variables, but it is unbounded for subWeibull variables because there is some convexity lost. The standard technique to prove the MGF failed for the heavy-tailed subWeibull random variables. This implies that if we do not study the MGF, we need to consider different decomposition on the general function $f-\mathbb{E}f$.
>
> To address the first challenge, we introduce Lemma 4, where we decompose the general function $f-\mathbb{E}f$ to the sum of independent sub-Weibul variables. In the proof of Lemma 4, a key step is that we need to construct a function $t \to h(\epsilon_n F_n(X_n, X'_n) +t)$ to apply our induction assumption. The proof of Lemma 4 may be simple, but Lemma 4 itself is very useful. For example, one can use Lemma 4 to prove more McDiarmid inequalities, e.g., the polynomially decaying random variables, which will enrich the family of McDiarmid inequality.
>
> To address the second challenge, rather than bounding the MGF, we bound the $p$-th moment of the sum of subWeibull random variables. Thanks to the fact that subWeibull random variables are log-convex for $\alpha \leq 1$ and log-concave for $\alpha \geq 1$, we can apply Lata{\l}a's inequality (Lemma 7 and Lemma 8) to bound this $p$-th moment. However, it is not a direct application of the Lata{\l}a's inequality. Lata{\l}a's inequality holds for the $p$-th moment of the \emph{symmetric} random variables. On one hand, we need to carefully construct new random variables to satisfy the symmetry condition, for example, we introduce random variables $Y_i$, $Z_i$ in the proof of Lemma 5. On the other hand, since we study the weighted summation, Khinchin-Kahane inequality is also required to use.
>
> Finally, we would like to quote the comment of Reviewer LoHq to confirm the technical novelties: "The proof consists of two steps: reduction to sum of independent sub-Weibul variables (Lemma 4), and the corresponding moment bound on a sum of sub-weibull rv's. The first part is done thanks to a conditioning trick with Jensen's inequality. The second part is done thanks to a carful application of Latala's inequality (Lemma 7 in the appendix)."
>
> Question 2: I would recommend moving the proofs of Lemma 3 and Lemma 4 to the appendix ...
>
> Answer: Thanks for your suggestion. We have moved the proofs of Lemma 3 and Lemma 4 to the appendix.
>
>
> Question 3: I might be missing something simple but I don't see ...
>
> Answer: In Lemma 4, a key Lemma to prove Theorem 1, we assumed $|S - S_i| \leq F_i(X_i, X_i')$ for $i=1,..., n$, where $S = f(X_1,...,X_{i-1},X_i,X_{i+1},...,X_n)$ and $S\_i = f(X\_1,...,X\_{i-1},X'\_i,X\_{i+1},...,X\_n)$, which can be analogized to the bounded difference condition $|S - S_i| \leq c\_i$ for $i=1...,n$. Here, $F_i$ is a arbitrary function satisfying the condition of subweibull random variable $||F\_i(X\_i, X'\_i)||_{\psi\_{\alpha}} \leq \infty$, which is unrelated to the distance. The result in Lemma 4 helped us went from a bound in terms of $f-\mathbb{E}f$ to a bound in terms of the $F\_i$. This argument can also refer to the comment of Reviewer LoHq: reduction to sum of independent sub-Weibul variables (Lemma 4).
>
> The assumption $|S - S_i| \leq F_i(X_i, X_i')$ can be seen as a Lipschitz condition when $F_i(X_i, X_i')$ is a distance function. For example, in Remark 1, by considering $F_i(X_i, X_i')$ as a distance function, we give concentration inequities in the context of the subweibull diameter.
>
> Question 4: Page 4: Population Risk
>
> Answer: Thanks for this review. We have fixed this typo.

---

> ### Author Response · Authors · 2023-11-22
> **Part II**
>
> Question 5: Issue in Weakness ``Though the generalization to unbounded losses is interesting, it would be nice if there are more applications presented ...''
>
> Answer: McDiarmid's inequality has been proved to be useful in a number of
> applications, such as algorithmic stability and suprema of empirical
> processes. This work extends McDiarmid's inequality to a broad class of heavy-tailed distributions and can motivate more McDiarmid's inequality by Lemma 4. Besides, proving generalization bounds for heavy-tailed losses is a significant direction in the learning theory community. For the tool of PAC-Bayes, Space Complexity, and Information theory, there are many works that study the heavy-tailed losses, while it is lacking for the tool of Algorithmic stability. Our work is the first attempt to give generalization bounds for the heavy-tailed loss for Algorithmic stability. Recently, (Bousquet et al. 2020) provided a sharper generalization bound for algorithmic stability. In their proof, they also use the $p$-th moment technique, and the $p$-th moment version of the Bounded differences/McDiarmid’s inequality is an important component. It seems to be straightforward to give better generalization inequality by combining our $p$-th moment inequality and the technique of peeling the dataset in (Bousquet et al. 2020). Since our work mainly follows the work  (Kontorovich, 2014; Maurer \& Pontil, 2021), we didn't consider giving sharper generalizations in the manuscript.
>  We will include this part of the results in the final version.
>
> References
>
> Olivier Bousquet and Andre Elisseeff. Stability and Generalization. 2002.
>
> Aryeh Kontorovich. Concentration in unbounded metric spaces and algorithmic stability. 2014.
>
> Vitaly Feldman and Jan Vondrak. High probability generalization bounds for uniformly stable algorithms with nearly optimal rate. 2019.
>
> Oliver Bousquet, Yegor Klochkov, and Nikita Zhivotovskiy. Sharper bounds for uniformly stable algorithms 2020.
>
> Andreas Maurer and Massimiliano Pontil. Concentration inequalities under sub-gaussian and sub-exponential conditions. 2021

---

### Meta-Review · Area_Chair_ASjL · 2023-12-12

**Metareview:**

The paper proposes a generalization bound for a variant of non-uniform stability condition. The analysis is based on a new form of McDiarmid inequality, where the differences are bounded by another function, which satisfies a sub-Weibul moment conditions. While the work is an interesting and technically non-trivial extension of existing concentration results the authors do not provide any new generalization bounds that would be of interest to a broader ML theory community. In addition, the result extend weak generalization bounds that have been strengthened significantly in a line of work starting with (Feldman and Vondrak, 2019) . Such stronger bounds tend to be crucial for getting tight generalization bounds.

**Justification For Why Not Higher Score:**

n/a

**Justification For Why Not Lower Score:**

n/a

---

### Decision · Program_Chairs · 2024-01-16

Reject